# The Current State, Challenges, and Opportunities of Recycling Plastics in Western Australia

Ana María Cáceres Ruiz * and Atiq Zaman

Curtin University Sustainability Policy Institute, School of Design and the Built Environment, Curtin University, Bentley, WA 6102, Australia
* Correspondence: 20097613@student.curtin.edu.au

**Abstract:** In 2018–2019, 85% of discarded plastics were landfilled in Australia. In Western Australia (WA), only 5.6% of plastics were recovered for reprocessing. With several Asian Countries imposing import restrictions, which were the prime destination for recyclables from Australia, the whole scenario for the waste industry has changed. Australia has now adopted export bans for recyclables, including plastics. WA is at a fork in the road; WA needs to rethink its relationship with plastic materials. This study explores how to create local markets for recycled plastics underpinning circular principles. The study examines barriers and drivers to enable markets for recycled plastics in WA through questionnaires, surveys, and interviews with relevant stakeholders. Poor source separation, low and inconsistent plastic waste feedstock, and virgin plastic competition are some of the challenges, while new investments in recycling infrastructure, WA's take-back scheme for beverage containers and circularity frameworks are drivers. This study concludes that a modulated fee-based product stewardship model focused on product design, along with strategies such as green procurement and landfill management modifications would promote a circular plastic waste economy in WA. This can create markets for secondary recycled plastics, minimize the over-reliance on fossil fuels and prevent plastics from leaking into ecosystems.

**Keywords:** recycling; plastic waste; market; drivers; WA; circularity; barriers; policies

## 1. Introduction

The management of plastic waste is one of the most challenging issues, as unmanaged plastics pose significant impacts to the environment [1,2]. The world crisis in managing plastic waste has intensified since the China Waste Ban in 2017. The ban restricts post-consumer plastic entering China from any overseas countries [3–5]. The legislation set stringent contamination thresholds of less than 0.5% [6], well below Australian contamination rate levels with averages between 6% and 10% [7]. To respond to the China Waste Ban, the Australian Government laid out legislation, through the Recycling and Waste Reduction Bill 2020, that regulates the export of unprocessed waste in line with the bans announced in 2019 by the then Council of Australian Governments (COAG) (COAG ban) [8]. From July 2021, unsorted mixed plastics can no longer be exported, and from July 2022, only single-polymer plastics that have been sorted and processed (e.g., flaked or pelletized) can be exported [9]. In view of Australia's export ban, the Australian Government launched the Recycling Modernization Fund (RMF) to boost local waste processing. The RMF supports new infrastructure investments to sort, recycle and remanufacture waste materials, including mixed plastics [10].

These measures are also coming into effect because similar to the poor plastics waste management worldwide, Australia is facing significant challenges in managing plastics waste. Firstly, landfilling is by far the preferred method of plastic disposal in Australia. In 2018–2019, 85% of discarded plastics was disposed of in landfills, nearly 13% was recycled and nearly 3% went into energy recovery [11]. Only 17% of plastics entering the waste

stream are collected for recovery and imported and locally manufactured plastics had low levels of recycled content, at only 4% [11]. Another worrying issue is the 130,000 tons of plastics leaking into Australia's natural environment every year [9,12]. Not only does plastic leakage contribute to greenhouse emissions, but it also handicaps the carbon fixation capacity of oceans [13]. Between several thousand and up to 40,000 pieces of plastic can be found per square kilometer in Australia. This equates to roughly three-quarters of all the litter along Australian coasts. This plastic debris is mostly generated from within the country and much of it is accumulated close to city areas [14]. This brings about dire consequences especially on marine biodiversity where numerous marine species have ingested plastics and been killed and wounded through entanglement. Incineration rates and consumption of single-use plastics have and will continue to increase due to the COVID-19 crisis [15,16]. The surge in disposable items because of the COVID-19 crisis includes disposable masks, gloves, and delivery packaging, among others. Similarly, the diversion of recyclables into landfills and increases in stockpiling have come about due to the slump in markets and the overdue processing capacity expansion [17].

There is also a gap in terms of research progress in plastic waste management in Australia [18]. While there have been studies on substitutes for virgin plastics, only a few recent studies look into types of plastic waste sources, current plastic recycling capabilities and plastic consumption issues [18]. Current research highlights that the plastic waste crisis in Australia will not be solved only by introducing bans, but rather by how effective governments, community and industries work together on regional, national, and global levels [18]. Additionally, it has been demonstrated that encouraging stewardship in coastal settings can lead to large-scale benefits, such as reduction in coastal plastic waste [19]. That said, more studies that further understanding of how plastic waste can be used as a resource to create markets and circular supply chains are needed in Australia.

Considering the crisis in managing plastics in Australia and the need for further research on plastic waste management in Australia, this study aims to identify potential drivers for promoting the development of waste markets for recycled plastics in WA, more specifically the Perth Metropolitan Area (PMA) under a circular economy framework. The article focuses on the following study objectives:

1. Exploring household plastics recycling practices;
2. Identifying socio-economic and technological barriers and drivers to creating a competitive market; and
3. Proposing policy recommendations to overcome those barriers in promoting end markets for recycled plastics.

## 2. Current Plastic Waste Management

In 2000, Australia consumed 1.5 million tonnes of plastic. In 2017–2018, the consumption almost increased three-fold, reaching 3.4 Mt. Of this consumption, 2 Mt (58%) were imported, and 1.1 Mt (32%) were locally sourced and manufactured from imported virgin plastic resins [17]. Only 0.2 Mt (6%) were locally sourced and manufactured from domestic virgin plastic resin and 0.125 Mt (4%) were locally sourced and manufactured from recycled plastic resin. A total of 2.5 Mt entered the waste stream, of which 2.2 Mt were landfilled [17]. About 0.32 Mt of plastic waste were recycled domestically and about half was exported.

The Australian plastic recovery rate in 2018–2019 increased to 11.5% (in 2017–2018 was 9.4%) mainly due to the growing energy recovery industry [18]. The main source of recovered plastics in Australia is municipal packaging (58%), and commercial and industrial (C and I) packaging (13%). Construction and demolition (C and D) waste, e-waste recycling and exporting second-hand clothes account for the remaining 29% of recovered plastics [18].

In Australia, the Australian Packaging Covenant (the Covenant) is the national, industry-driven packaging stewardship scheme. The Australian Packaging Covenant Organisation (APCO), an independent, not-for-profit entity manages the Covenant to ensure the compliance of Australia's 2025 National Packaging Targets (2025 Targets). The targets set a guide to achieving a new sustainable outlook for packaging and apply to all

packaging made, used and sold in Australia [20]. Despite its importance, SWOT analysis has shown that due to factors such as financial contribution commitments and enhanced reporting, many companies feel discouraged from participating in the Covenant and may opt out of it altogether [18]. Those businesses who choose not to become a signatory of the Covenant or that want to withdraw must meet the obligations by states and territories under the National Environment Protection (Used Packaging Materials) Measure 2011 (NEPM). However, an independent review of the NEPM found no evidence of reported actions, complaints or investigations from states and territories in four years [21].

In terms of public awareness, APCO developed the Australasian Recycling Label (ARL), an on-pack labelling system to help consumers make the correct recycling choice [22]. See Figure 1. In Australia, a kerbside collection system allows for the collection of mixed recyclables, including eligible plastic waste (e.g., plastic bottles), in co-mingled recycling bins. Products such as plastic bags, foam, multi-layered plastics, and bio-based (compostable) plastics are considered contamination when placed in the yellow top bin (recycling bin). The recycling bin contents are transported by councils or contractors to Material Recovery Facilities (MRFs), where the contents are sorted and baled.

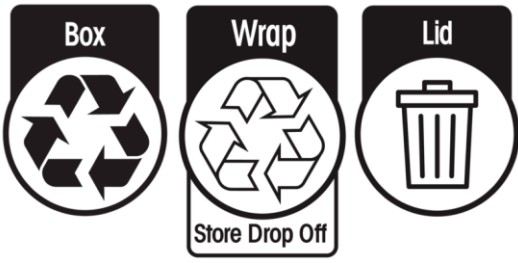

**Figure 1.** The Australasian Recycling Label (ARL). Planet Ark, PREP Design & Australian Packaging Covenant Organisation, CC BY-SA 2.5. [23].

As for WA, the 2018–2019 Australian Plastics Recycling Survey revealed that WA consumed, for the period 2018–2019, a total of 354,500 plastic tonnes [24]. Only 20,000 tonnes were recovered, meaning that the recovery rate of plastics in WA was 5.6%, well below the national rate, at 11.5%.

In early 2019, the WA government launched the Western Australian Waste Avoidance and Resource Recovery Strategy 2030 [25]. The objectives of the strategy revolve around the avoidance and recovery of waste and protection from the impacts of waste. The strategy has emphasised a list of eight focus materials as materials that are priority in terms of actions and measurement. Plastic packaging and plastic containers are part of the list as they make up for a considerable share in the waste stream [25]. The strategy underlines that disposing of plastic in landfills is an economic loss and that it negatively impacts the environment [25]. The strategy acknowledges that plastics are a high-value commodity subjected to low contamination levels [25].

*Circularity in Plastic Waste Management*

The literature review of current plastics of plastic waste management indicates an urgent need for holistic strategies (such as a Circular Economy (CE)) and tools (such as an Extended Producer Responsibility (EPR)) in creating an end-market for recycled plastics. These have been developed as policy measures to manage plastic waste and to reduce leakages of plastic debris into the natural environment [26,27].

EPR schemes address what is considered to be the weakest link in the value chain: how products are disposed of in the post-consumer phase [28]. Because with EPR, producers are held accountable for a significant part of negative externalities brought about by the final disposal of their products, they are pressured to drive upstream changes, namely, in materials and the design of products [28,29]. These changes will eventually help increase the use of recycled materials in production, coupled with increased resource efficiency [28]. Producers can fulfill individually or collectively their EPR commitments. The former

applies the "polluter pays principle", and the latter model contributes to a Producer Responsibility Organisation (PRO) by paying an EPR fee [29]. Experiences with EPR programs in various European Union (EU) countries have demonstrated that the concept can be key when moving towards a CE [30].

Nevertheless, despite the extensive use of these schemes, and even though the EPR schemes certainly provides incentives for virgin material substitution, no major changes have been identified in regard to improving the design for recyclability. In fact, there is not much evidence that shows that these schemes have had an impact on product design [29]. In the Nordics, for example, EPR schemes were introduced in the 1990s, especially for packaging waste streams, to design out waste and to foster recyclability. The recycling rates did increase but product design remained unaltered [31].

In recent years, several countries such as France, the Netherlands, Italy, and Sweden have started to modulate EPR fees considering measurable environmental characteristics of a product. These include recyclability, reusability, recycle content, polymer of choice, type of additives, end-markets, availability of technologies to reprocess the plastic material, etc. [29,32]. There are basic and advanced types of eco-modulation of fees. Basic approaches involve material weight and type of polymer considering end-of-life cost differences [29] and in turn other aspects of eco-design are overlooked, namely, recycled content and recyclability [32]. Advanced fee eco-modulation operates by rewarding producers, through "bonus" adjustments, or penalising them through "malus" adjustments, thus creating a system of differentiated fees. Should fees be differentiated enough to have an impact on producers' costs, they will be more inclined towards more affordable, sustainable options [32]. In theory, the more sophisticated the EPR fee modulation is, the more that the Design for environment (DfE) of products to reduce environmental impacts is instigated [29]. Advanced eco-modulated fees have only recently been applied; therefore, there is not much information on their performance. Some initial results from the French EPR scheme for packaging show that the number of products with malus penalties reduced over time [29].

## 3. Materials and Methods

### 3.1. Theoretical Framework

This research follows a system thinking approach in which exploration is carried out by examining each element of the system, and the interactions and linkages between them will set the frame for this study. Systems thinking aims to understand systems in a holistic way, not in isolation, and is particularly useful in tackling wicked problems, such as environmental degradation [33,34].

CE principles underpin the outcomes of this research. Kirchheer et al. (2017) define CE as an economic system that replaces the end-of-life concept by reducing, reusing, recycling and recovering materials (4R) at various levels, including companies, products and consumers as well as other bigger spheres, such as cities and nations [35]. The overarching aim of CE is to achieve sustainable development [35], CE aims at creating social, economic and environmental value [36]. It is worth pointing out that the "Recover" dimension is not deemed to be a long-term solution that supports a circular economy [37] and as such this study will not focus on this dimension.

### 3.2. Study Design

This study used a mixed methodology by combining both qualitative and quantitative analysis. Several approaches to collect data were applied in the study. Firstly, an online questionnaire to members of the public was developed to address research objective 1. Secondly, for objectives 2 and 3, an online survey was conducted to local re-processors and an in-depth, semi-structured interview with experts. The data obtained were both quantitative and qualitative. Both the questionnaire and the survey were developed using the Qualtrics survey software package. The questionnaire link was shared with

the community using social networking services. The survey link and the invitation to interview was conducted through LinkedIn and/or business email.

Due to the population distribution in WA, most of the public that participated in this study belonged to Perth and Peel regions. Only a low percentage were from regional areas in WA. The study also included some visits to plastic reprocessing facilities, having interviews with workers, and seeing first-hand the process. The participation of members of the public in this study was anonymous and there were not duplicate IP addresses across responses. Additionally, the data protection in this study does not permit publishing the name of the interviewees, including plastic recyclers.

### 3.2.1. Community Questionnaire

To understand people's attitudes and behaviours towards recycling an online questionnaire with mostly rating scale questions was designed. Information such as age band, postcode, income bracket, and number of waste bins was gathered to see if there was a correlation with demographics or type of waste collection and attitudes towards recycling. Rating scales are widely used in research to help obtain a measure of attitudes [38]. Likert questions included:

- The Australasian Recycling Label (ARL) has changed recently. How easily do you understand the label information?
- Does the new ARL help to recycle appropriately?
- How often do you put the following plastic items in the recycling bin? (Full list of items in Appendix B)
- How often do you take your soft plastic packaging to the shops where you bought the products?
- How often do you see if the plastic products/packaging are made from recycled materials?
- How likely would it be for you to buy products manufactured from recycled plastics over virgin plastics available on the market if the function and quality are the same?
- How likely would you be buying products manufactured from recycled plastics over virgin plastics available on market if the price is slightly higher?
- One of the biggest challenges of marine pollution is due to the plastic waste we generate every day. Do you think you are doing your best in recycling correctly to save the planet and reduce the pollution?
- If currently you are not doing your best, would you be doing any of the activities in the future to make a difference?

The community was also asked, with an open-ended question, if they have any suggestions on how to improve our correct recycling practice.

A tally of 335 people completed the online questionnaire, where the distribution of respondents is homogeneous, see Appendix A. Almost all of them reported living in Perth and Peel regions (97%). The last census in 2021 indicates that WA has 2.7 million inhabitants and Greater Perth has 2.1 million (84%) [39].

Most people fall into the Millennials age group (64%), followed by people from generation X (41–56 years), at almost 19%. Most people fit into the secondary and tertiary education level, at about 28% and 34%, respectively. Half of the population fell into the middle-income bracket. Further detail on demographics can be seen in Appendix B.

The use of self-reported data has some downsides. Self-completed questionnaires are subject to response bias, sampling bias or the participants may not assess themselves accurately. However, given the limited sources of other independent data on the topic available at the time of the research, the importance of collecting these data outweighed these limitations.

### 3.2.2. Reprocessors Survey

The survey was designed to understand plastic recyclers views on recycling plastic markets in WA. Four reprocessors completed the survey. Two from current recyclers (one of them being an e-waste recycler who also deals with some plastic polymers) and two

from the RMF's successful recipients and whose facilities are not yet in operation. The table below shows the polymers that they deal with.

Questions included:

- Which plastic wastes are processed by your facility?
- What are the sources of plastic waste processed at your facility?
- What are the end-uses of your processed plastics?
- What are the main challenges that your facility faces in terms of recycling plastic waste?
- Based on WA's local context, please rate the level of recyclability and cost-effective of plastics waste from very easy and very cost effective to very difficult and very costly to recycle.
- What are the biggest challenges in creating markets for recycled plastics in WA?
- How to overcome the challenges (that you mentioned before) for creating a local market for recycled plastics in WA?

The low response from the plastic reprocessors group to the survey is compounded by the fact that there are not many plastic recyclers currently operating in WA, which are five according to the latest report data [40]. During the data collection stage, it was found that now there are four plastic reprocessors, as one of them ceased operations in 2020. Hence, the survey link was sent to four of the reprocessors currently operating and to all RMF recipients in WA (three). Plastic recyclers obtain the feedstock from a variety of sources. The highest share (just shy of 24%) is observed for local government collection facilities (MRFs). Quite interestingly, the proportion was the same (at almost 18%) for local government collection facilities (recycling drop-off points), Containers for Change (a Deposit Refund Scheme for beverage containers) drop-off points, commercial/institutional sources, and industrial sources. The other sort of source chosen, about 6%, was specified as general public. None of the recyclers reported receiving feedstock from other Australian states. Regarding the end-use of their recyclates, an equal share of 38% for both manufacturing in WA and exporting was found. Using recyclates within Australia for manufacturing and sending to other facilities in Australia for further processing are both at 13%.

3.2.3. Experts' Interviews

As this study looked to gain a deep understanding of experts' views about creating markets for plastics recycling, conducting interviews was deemed to be the most fit-for-purpose method. Interviews are a suitable method for collecting people's insights and experiences [41]. The interviews followed a semi-structured approach, which lends itself to having an ordered but flexible questioning [42]. The experts were selected based on their association with plastic waste management entities and organizations, and their involvement with the policymaking of plastic-waste-management-related issues. Thus, the participants of the experts' interviews belonged to three main groups:

i.   Government organizations, including local and central;
ii.  Business organisations, such as plastics recyclers; and
iii. Environmental organisations, including waste management services providers and non-governmental organisations.

Appendix C shows further details on each expert participant.

A total of 23 invitations were sent and, ultimately, nine interviews were conducted (participation rate: 39%). The waste expertise of participants was on average 12.6 years. The interviews with the experts were audio-recorded; this allowed for a more natural flow of conversation [42].

Questions included:

- Based on your knowledge, what are the key challenges and opportunities of recycling plastics waste?
- What actions can be undertaken to increase the current plastics recovery rate in WA (5.6%)?
- In terms of the market of recycled plastics, what are the key reasons for Western Australia not creating favourable end-markets for recycled plastics?

- Do you think that the geolocation of Perth and WA is an advantage or disadvantage when it comes to creating new markets for recycled plastics?
- Virgin plastics are often preferred raw materials due to aesthetics, concerns about quality and low-cost reasons. How to create more competitive markets for recycled plastics under this market condition in WA?
- APCO has laid out a set of minimal recycled content requirements for packaging, but they are voluntary targets. Do you think that these should be mandatory requirements?
- To achieve APCO/National Packaging Targets, what do we need to do?
- Markets for poor quality mixed polymers are very low, while single polymers can be reprocessed more easily and have more markets. How to incentivise industries to manufacture plastic items with only one type of plastic resin and with high recyclable plastics such as PET, HDPE, and PP (not black)?

### 3.3. Data Analysis

The Likert scale data were analysed using graphical visualisation and tabular representation to identify trends. Analysis data from different questions were also analysed on Qualtrics, using the Breakout tool in the Advanced-Reports Visualisation section. Additionally, to correlate different (demographic) variables and the data collected, the "Relate" analysis option was used. This option helps determine the statistical relationship (running a Chi-squared test) among variables of interest.

The qualitative data obtained from the open-ended questions were analysed to identify themes and associations. The recordings from the interviews were transcribed through the speech recognition software from Microsoft Word. The transcripts were read through to search for themes. This search is about identifying the underlying meaning of what has been said and is a form of coding [42].

## 4. Results and Discussion
### 4.1. Community Questionnaire Results
#### 4.1.1. The On-Pack ARL

Most participants find the ARL both easy to understand and (very) helpful to recycle appropriately. About a quarter think the label is either neutral or moderately difficult to understand. Only ~1% found the ARL to be very difficult to understand. About a third think the label is either neutral (17%), slightly helpful (10%) and not helpful (8%) when it comes to advising on recycling, see Figure 2.

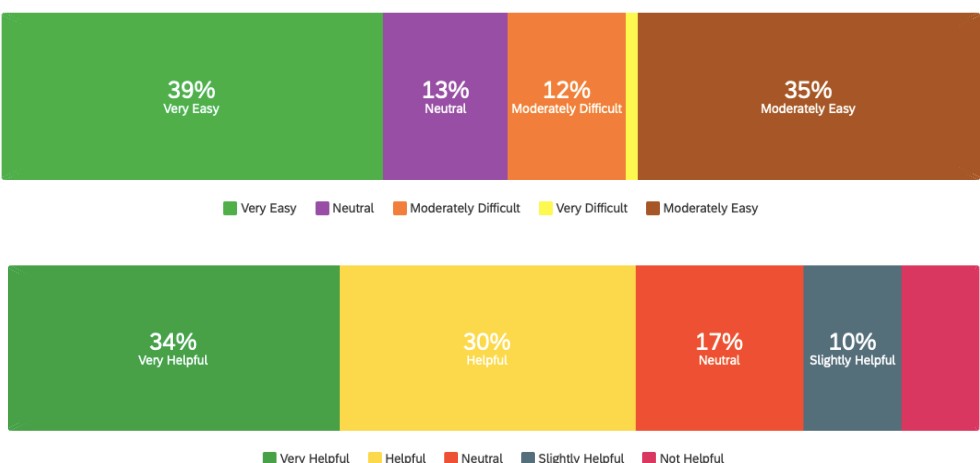

**Figure 2.** Above, the Australasian Recycling Label (ARL) has changed recently. How easily do you understand the label information? Below, does the new ARL help to recycle appropriately?

4.1.2. Source Separation

There is a gap disconnect between what people are placing in the recycling bin and what they should be placing in them. For example, plastic lids and biodegradable/compostable items are the items that people most misplace, at ~83% and ~50%, respectively. Additionally, only ~70% of plastic bottles/containers emptied and rinsed are always placed in the recycling bin. See Table 1.

**Table 1.** Survey's participants.

| Plastic Recycler | Polymers |
|---|---|
| K | HDPE, PP, EPS, mixed plastics (only lids) |
| L | PS |
| M | PET, HDPE |
| N | PET, HDPE, PP, mixed plastics |

Plastic bags or soft plastics are problematic for the community. Firstly, about 26% of respondents wrongly placed them in the recycling bin. Secondly, half of people never take their soft plastics to the dedicated drop-off points, about 20% of participants rarely or sometimes do, and the remaining participants who always or very often return these plastics to the shops are only about 16% and 12%, respectively.

4.1.3. Choice of Recycled Products

Generally, people do not tend to identify if the products/packaging they buy is recycled. There was not a noticeable trend as to how many people make sure that the products they buy have or are from recycled materials, with the highest share falling into the category of "sometimes I see if the plastic products/packaging are made from recycled materials" at ~27%.

4.1.4. Occupation Status and Choice of Recycled Products

There is a statistically significant relationship between the occupational level and the choice of recycled products that are slightly higher in price ($p = 0.0271 < 0.05$). The same occurs when comparing this purchasing option with the income bracket ($p = 0.0314 < 0.05$). Then, it can be argued that money plays a key role in consumer buying preferences when the price is higher. This means that the more people earn, the more likely they are working full time, and so are the chances to buy a slightly more expensive recycled plastic product. In other words, wealthier people are more prone to choose recycled products regardless of the price being altered. As a matter of fact, the full-time employees' group, the highest earners, was the only group inclined to buy these products; the other occupational groups were not inclined to buy them. See Appendix D.

The occupational status and income bracket were also tested to see if they were statistically significant to the variable "How likely would it be for you to buy products manufactured from recycled plastics over virgin plastics available on the market if the function and quality are the same?", which does not mention price. However, in both cases $p > 0.05$, meaning that these two demographic variables were not statistically significant to buying recycled products with good quality and function as virgin plastics. In other words, money is not a factor when opting for recycled plastics rather than virgin plastics so long there are no changes in prices. If there is a price alteration, then money becomes a constraint. This supports the findings laid out above.

4.1.5. Willingness to Buy Recycled Products

People are inclined to buy recycled products for environmental reasons. People are 57% "highly likely" to buy recycled products over products made from virgin materials as they "care about the environment". Nearly 30% are likely, as it serves the same purpose,



and 10% are neutral. No one picked highly unlikely. The pie charts (Figure 3) show the breakdown of answers.

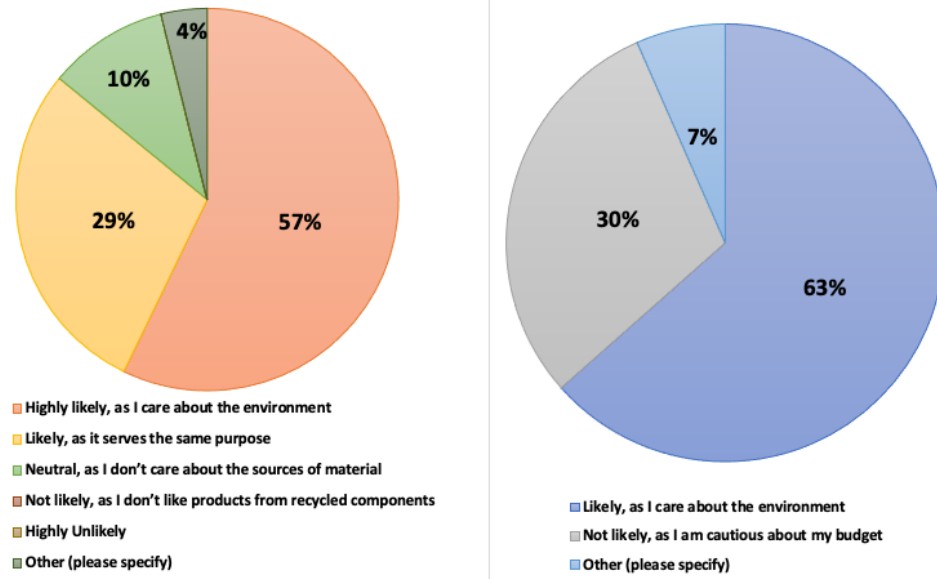

**Figure 3.** On the left, "How likely would it be for you to buy products manufactured from recycled plastics over virgin plastics available on the market (e.g., storage container, bottles, etc.) if the function and quality are the same?". On the right, "How likely are you to buy products manufactured from recycled plastics over virgin plastics available on the market (e.g., storage container, bottles, etc.) if the price is slightly higher?".

People are generally confident in their recycling choices. They were asked "One of the biggest challenges of marine pollution is due to the plastic waste we generate every day. Do you think you are doing your best in recycling correctly to save the planet and reduce pollution?". About 66% of respondents said that they often recycle correctly. Second, ~15% answered that they always do it right and only a few people said that they seldom and rarely recycled properly, at ~9% and ~2% each.

In general, people are keen to make a difference. People were asked about any activity they might be engaging in to make a difference if they happened not to be doing their best in sorting their waste. 58.5% of respondents are keen to participate in future activities to make a difference (e.g., put more effort and time to recycle correctly; check on the council's website to know which items I can recycle). Nearly 22.5% of people will not change their habits to recycle better even though they are not doing their best. Roughly 19% of respondents claimed they would do something else, for example, "Try not to buy anything involving plastic" or "Try to bulk purchase dry foods with reusable containers.

4.1.6. Perception of Recycling Practices

Self-perception of recycling habits are significantly influenced by demographic factors, including age and number of bins. Other factors, such as education, income, occupational status, and household size, have no influence. Further details below.

- There is a statistically significant relationship between the variable "Which age band do you fit into? "and "One of the biggest challenges of marine pollution is due to the plastic waste we generate every day. Do you think you are doing your best in recycling correctly to save the planet and reduce the pollution?" ($p = 0.000147 < 0.05$). Most Boomers (57–75 years) and Post War (+76 years) tend to say that they always recycle correctly, while the rest of the generations, Gen Z (25 years), Millennials (25–40 years) and Gen X (41–56 years) tend to go for the I-often-recycle-correctly option. On the

whole, the older the person, the higher the confidence in their recycling habits. Thus, the perception of recycling habits is significantly influenced by age.

- There is also a statistically significant relationship between the variable "How many waste bins does your household have?" and "One of the biggest challenges of marine pollution is due to the plastic waste we generate every day. Do you think you are doing your best in recycling correctly to save the planet and reduce the pollution?" ($p = 0.000178 < 0.05$). The trend identified is that having a two-bin system makes people say they are not that good at recycling, while having three bins makes them think they are doing it correctly. For example, most people who say that they always recycle correctly have a three-bin system. In contrast, the majority of people who said that they seldom recycle correctly have a two-bin system. See Appendix E.
- The other two variables found to be statistically significant (in most cases) were "Do you think you recycle correctly" and "how often do you recycle these plastic items?". Generally, while people tend to recycle some items well, some other items tend not to be recycled properly. This means that some items generate more confusion than others, namely, throwaway coffee cups, biodegradable plastics, foam, and squeeze packs. Similarly, some people are not recycling as well as they think. Even in the best-case scenario, when people tend to recycle properly, there is still room for improvement. For example, the best percentage to correct answers was 70% for people always disposing of rinsed/empty plastic bottles in the recycling bin. It is worth mentioning that although $p < 0.05$ in the case of plastic lids, only 16% of people chose the right answer. See Table 2.

**Table 2.** *p*-value between "Do you think you recycle correctly" and "how often do you recycle these plastic items?", and correct responses to the question "How often do you put the following plastic items in the recycling bin?".

| *p*-Value < 0.05? | Item | %Correct Answers |
|---|---|---|
| Yes | Plastic bottles/containers, rigid plastic containers (empty/rinsed) | 70% |
| Yes | Bottles and containers tops/lids | 16% |
| No | Takeaway coffee cups | 41% |
| Yes | Plastics bottles/containers with liquid contents | 62% |
| Yes | Plastics bags | 69% |
| Yes | Recyclables plastics items inside a plastic bag | 73% |
| Yes | Plastic straws | 52% |
| Yes | Plastic wrappers (including bubble wrappers) | 65% |
| Yes | Balloons | 63% |
| No | Foam (e.g., take out/away containers) | 63% |
| No | Squeezy packs (e.g., baby food package, toothpaste tubes, etc.) | 75% |
| No | Biodegradable and compostable plastics | 41% |
| Yes | Bicycle/car's Tyres | 71% |

### 4.1.7. Themes Identified from the Community Questionnaire

The following themes were identified from the question "Do you have any suggestions on how to improve our correct recycling practice?"

- Bin-systems

Commingled bins should be reconsidered. Separating out by material type (glass, metal, plastics) can reduce the contamination of recyclables. For example, many people claimed that taking soft plastics back to the store is too much of a hassle and many called for a dedicated soft plastics bin as a part of the kerbside system. This explains why only 16% of people always returned the soft plastics to store drop-offs. This is a big issue, considering that 33% of plastic packaging is soft plastic. Separating out soft plastics can help in the recovery of these materials and reduce contamination of other waste streams.

- Collection Practices

Household waste collection frequencies should also be reconsidered. The general (rubbish) bin is collected weekly and the recyclables bin fortnightly. Recyclables are bulky and take up more space than rubbish items. This means that the recycling bin will fill up before the rubbish bin, wreaking havoc on sorting rates at a household level and in turn, in plastic collection rates. In fact, many people admitted putting their recyclable waste in the rubbish bin because the recycling bin was full even when knowing that that was not the place for recyclables.

- Distrust in the system

Some people see recycling as a waste of time because they think that their waste ultimately goes to landfills. As there is no clarity in what happens with the product thrown away, people do not bother and end up not feeling responsible for their waste. It is imperative to launch marketing/education campaigns to show citizens what happens with their waste after it gets picked up by the council. Awareness and education of waste are key drivers of behaviour change [43].

### 4.2. Identified Key Barrier Themes from Interviews to Experts and Surveys to Reprocessors

#### 4.2.1. Feedstock: Insufficient and Inconsistent

It is a struggle for plastic reprocessors to find enough supply to keep up with local manufacturers. To make matters worse, the supply of material is conducted through a tendering process that generates feedstock variability. WA recyclers struggle to source enough material. In addition, the lack of volume hinders the commercial viability of operating a plastic reprocessing facility, and this can be an entry barrier for new players. Unless there is a commercial agreement to ensure supply, such as the deal for the new RMF funded facilities, there will not be a business case for new entrants.

#### 4.2.2. Polymer Types 3–7: A Sticking Point

Mixed plastics, which are plastics not of a single polymer, are no longer permitted to be exported from Australia. There are two types of mixed plastics. Those that include polymer 1 and 2 (mixed 1–2) and those that do not (mixed 3–7). Whilst 1–2 mixed plastics have a low value per tonne, 3–7 mixed plastics, or the "3 through 7" bale, are costly to separate and valueless (−$20 per tonne), and the likelihood of being deposited in landfills is high [17]. This has represented a massive burden on the waste management system in Australia.

Although some companies in Australia have been able to develop products using mixed plastics 3–7 (e.g., in outdoor furniture) [17,44], the niche markets for these products were saturated even before the export ban started. With the mixed plastic export ban already in effect, mixed plastics supply has increased even more. Due to the limited and saturated end-markets, these plastics might end up in landfills [17]. This issue is exacerbated by the fact that landfill levies in WA are one of the lowest national levels [11]. Additionally, landfilling in rural WA is more economical than landfilling in the city. Therefore, currently, for mixed plastics in WA, landfilling makes more economical sense than recycling. This supports the literature where lower cost of alternate methods of disposal have been identified as a key barrier to the commercial viability of recycling [45].

On the other hand, single polymers 1 and 2, PTE and HDPE, are the most coveted polymer resin materials due to their commercial value. Polymers 1 and 2 are highly recyclable and do not represent high material losses. This is also true in WA. When plastic recyclers were asked "Based on WA's local context, please rate the level of recyclability and cost-effectiveness of plastic waste from very easy and very cost-effective to very difficult and very costly to recycle". Plastic polymers 1, 2, and 5 fall into the (very) easy and (highly) cost-effective range, while plastic polymers 3–7 (not 5) fall into the (very) difficult and (very) costly band. See bar chart in Appendix F.

#### 4.2.3. Virgin Plastic Competition

All participants agreed that low-cost virgin plastic was a barrier for recycled plastics markets. There are many factors that contribute to this. Firstly, impure (high-contaminated)

recovered plastic feeds increase sorting and processing cost, and this hampers the ability to compete with cheap virgin resin. Secondly, the price of oil and gas is going down. If the price of virgin materials decreases, so does the price of recycled materials. Recycling plastics businesses (and markets) are very vulnerable to virgin market fluctuations because of increased financial risks (CIE, 2020). Thirdly, future projections set plastics as the main product of the oil industry. For example, British Petroleum has forecast that plastics will make up 95% of the oil demand growth by 2040 [46]. Lastly, the plastic recycling industry is a very commoditized industry. There are many brokers (middlemen) who aim to find the highest price for the recyclable. Commoditizing recyclables then increases the costs for plastic reprocessors, which in turn hinders their ability to find competitive prices to be able to stack up against fossil-based materials. However, a shift is now taking place with the COAG ban, especially in regard to export markets.

### 4.2.4. Challenges in Current Plastic Products Characteristics

Participants argued that the design of plastics products should be aimed at improving their management throughout their lifecycle. This includes reassessing characteristics that could affect recyclability properties, for example, reconsidering the need for composite plastics in products, reassessing the use of multiple types of polymers in a product, and the use of hard-to-recycle plastic materials given WA current infrastructure capabilities, namely, dark-colour plastics or plastics with hazardous substances. Another key factor to assess during the design stage is the end-markets of the recyclates, even for those materials that are not currently being processed; currently, there is low demand for recycled plastics in Australia. Finally, the use of plastics in the first place must be reassessed, as there is a considerable proliferation of unnecessary plastic packaging in the market.

### 4.3. Identified key Driver Themes from Interviews to Experts and Surveys to Reprocessors

#### 4.3.1. Infrastructure

Investing in new recycling infrastructure has been identified as an opportunity to promote a circular economy in the recycling stage by Australia's National Science Agency [44]. On this note, through the RMF, the state and federal governments have granted funding to three major projects that will be building infrastructure to expand the plastic recycling capacity in WA.

Two of the projects will be dealing with post-consumer plastic packaging, specifically with PET, HDPE, LDPE, and PP. This is key considering that, in Australia, these polymers are predominantly used in packaging consumer applications: ~42% of PET, ~37% of HDPE, ~48% of LDPE and ~33% of PP [40].

It is worth mentioning that just as fostering the plastic recycling infrastructure to process inland materials that were previously exported, investments in advanced separation technology in MRFs are needed. This would increase sorting efficiencies, reduce contamination, and sort more polymer types, including PP, PVC, and PS [44]. Landfilling plastics would be reduced, and local recyclers would be able to receive the much-needed plastic feedstock.

#### 4.3.2. Deposit Refund Scheme (DRS)

The Containers for Change program, a deposit refund scheme for beverage containers launched in WA in October 2020, has increased the number of recyclables that recyclers receive and reduced contamination levels. This is supported in the literature where it was found that, with DRS or container deposit schemes (CDS) in place, the number of recovered containers per capita increased [47].

#### 4.3.3. Circularity

All participants agreed that circularity should be the frame of work to enable local markets for recycled plastics. A participant clearly outlined it: "we should avoid generating waste in the first place and we should go more to a circular economy concept" and then

added "circular economy is two things, the circularity of materials and the economic opportunity." This vision is supported by the literature; the Ellen MacArthur Foundation named it "The New Plastics Economy" [48]. In this new economy, plastics never become waste, do not leak into natural systems, and are decoupled from fossil feedstocks [48]. The cornerstone of this new economic model is to come up with an after-use plastics economy that captures material value, increases resource productivity, and provides incentives to avoid plastic leakages [48]. One of the strategies to realise this new economy is to increase the uptake of high-quality recycling significantly. The Ellen MacArthur Foundation also recognises, like all interviewees did, that this, of course, implies keeping plastics in the system because that is the best way to reduce leakages of plastics into the natural environment and other negative externalities [48]. A circularity approach for plastics decouples the reliance on fossil fuel and prevents plastics from leaking and becoming toxic to the environment.

### *4.4. Policy Recommendations*

#### 4.4.1. The Modulated Fee-Based EPR Scheme

To overcome identified barriers and make the most of the drivers to enable local markets for recycled plastics, almost all experts agreed that the economy must be pushed towards circularity by implementing an Extended Producer Responsibility (EPR) scheme for plastic packaging. Plastic packaging must be brought under the national legislation for product stewardship as a mandatory scheme that underscores eco-design. Improving design will lead to reduced disposal and increased recycling rates. All experts agree that the EPR scheme should be mandatory. This situation was clearly described by a participant: "It should be mandatory to be a member of APCO and you should have to achieve those targets rather than the current voluntary-ish, coregulatory-ish approach . . . " and also added on the same topic, "I think we've had enough carrot like there has been carrot for 15 plus years".

The experts said that we should be drawing on international experiences, particularly the EU. In the EU, all packaging EPR schemes have some very basic fee modulation. At state and national levels, the APCO 2025 Targets could serve as eco-criteria. These targets are key to creating local markets for recycled plastic. A participant noted about recycled plastics in WA, "The markets will probably still be export markets in large unless there's a large uptake of recycled content requirements in Australia". Table 3 gives an overview of the types of fees that could be applied given the 2025 Targets.

#### 4.4.2. Sustainable Procurement

In 2018–2019, about 60% of plastics consumed in Australia were imported. Around 36% came from local manufacturers using local or imported virgin resins and only 4% were recycled plastics [40]. One of the key actions of The National Waste Policy Action Plan 2019 is to use the Commonwealth's purchasing power to increase demand for more sustainable goods, including recycled plastics. Most participants strongly agreed that the mass application and consideration of sustainable procurement strategies (e.g., purchasing products with recycled contents) would create local demand for recycled plastics products.

#### 4.4.3. Landfills Management

To achieve a plastic circular economy in WA, it is necessary to revise landfills' management. Currently, landfilling makes more economic sense than other options to recover plastic materials. Experts agreed that landfill levies should be increased. One of the barriers to increasing recycling rates in the low cost of other disposal methods [45]. The 2019–2020 waste landfill levy was AUD 70/ton in Perth. This is a low rate. In Metro Adelaide, it was AUD 110–AUD 140/ton, and in the NSW metro area the levy was 143.60/ton. Another key issue is the open licence that mining companies in WA have for burying dewatering HDPE pipes, highly recyclable materials, in their landfills. This represents a big opportunity in the

market for WA. For example, there has been a reported usage of 8000 tonnes of dewatering pipe per year by one.

**Table 3.** EPR modulated fees using the APCO 2025 Targets as "eco-criteria".

| APCO 2025 Targets "Eco-Criteria" | Type of Eco-Modulated Fee |
| --- | --- |
| 70% of plastic packaging recycled or composted<br>100% of packaging to be reusable, recyclable, or compostable. | Format design: Different elements on the same product (glues, ink, sleeves, valves, caps) affect sortability and recyclability.<br>Available technology to sort/reprocess the plastic product: Fees should be based on the existing sorting and recycling infrastructure.<br>Reusability: If plastic packaging can be reused, some countries, such as the Czech Republic have decided not to charge any fee. This type of fee is still nascent though it has not been adopted widely [32].<br>Polymer of choice: The processability level determines this fee. If the product is easy to recycle (polymers 1 or 2). This fee is crucial to ensure high-grade recycling products, where the materials retain their value and quality; therefore, no material degradation occurs [48,49]. |
| Phase-out problematic and unnecessary single-use plastic packaging | Disruptive additives: Opacifiers and black pigments are considered disruptive because they prevent the plastic from being identified as one at the sorting stage (at the MRFs).<br>End-markets: Fees dependent on whether the recyclates of the plastic product have an entry as secondary raw material [32].<br>Reduction priority items: Items should have higher fees if they are part of the priority items that the Government is trying to phase out, namely, multi-material laminate, soft plastics, composites, EPS, opaque PET, rigid plastic with carbon black, PVC packaging [50]. |
| 50% average recycled content across all packaging<br>>20% for all plastic packaging | Recycled content per polymer type: For example, by material-specific recycled target of APCO (e.g., PET 30%, HDPE 20%, PP 20% and flexible plastics 10%). |

## 5. Conclusions and Further Research

The China ban generated waste turmoil in the world and, as a result, many countries are trying to readjust to the new waste market outlook. This research set out to explore how to enable local markets for recycled plastics in WA, where less than 6% of discarded plastics are recovered from being reprocessed. To this purpose, this research sought to identify barriers and drivers that could foster the development of markets for these materials. The theoretical framework for this research was based on circularity principles, including reducing, reusing, and high-grade recycling.

A questionnaire to the community was developed to understand their attitudes and habits towards recycling. In large, despite most people claiming that they recycle adequately, this was not always the case. There are some items that people generally sort properly, while others seem to generate confusion. Generally, people are confused about how to recycle, which is exacerbated by the collective distrust in the waste management system. Hence, there is a need for greater environmental awareness and education.

Self-perception of recycling habits was found to be significantly influenced by factors such as age and number of household waste bins. In addition, most people with higher incomes tend not to mind buying recycled products with a higher price than virgin products; if the price is not affected, most would opt for purchasing recycled products regardless of their income.

The most prominent barriers to create circular plastic economies in WA identified were plastic waste feedstock shortage, competition with virgin resins, poor source separation at households, and saturated markets for polymers 3–7. To overcome these challenges, several measures are proposed. Firstly, implementing a mandatory EPR scheme for plastic packaging based on the APCO 2025 Targets with modulated fees. The scheme must focus on eco-design and international experiences show that eco-modulating fees can foster environmentally friendly designs. This could reduce landfilling and improve recycling rates as well as recycled content in products. Drivers such as new infrastructure, through

the RMF, and Container for Change can support the scheme. Additional measures include green procurement strategies and reassessing the landfill management strategy in WA.

In large, an unmanaged plastic waste value chain poses a serious threat to the environment. Waste is the downstream of all people participating in the value chain. Therefore, it all comes down to how to best manage people who are part of each stage of the value chain. Designers, manufacturers, distributors, consumers, and governments are part of the puzzle and must be held accountable. The proposed measures present an option to improve the management of the plastic waste value chain while reducing the overreliance on fossil feedstocks and enabling high-grade recycling markets.

For further research, there is an opportunity in improving plastic waste collection. The plastics collection efficiency in Australia is only 17%, in other words, only 17% of the plastics that enter the waste stream reach the sorting stage. This is a key factor as to why recyclers in WA struggle to find material feedstock. There are various possible reasons for this situation. Firstly, commingled kerbside collections are prone to high levels of contamination. Secondly, the collection frequency is low, and recyclables end up in the general waste stream. What changes/actions are needed to be conducted to increase the collection efficiency? Arguably, the recommendations herein proposed can contribute to this effect. Nevertheless, associated costs with increasing frequency collections, for example, must be assessed. Will more and cleaner waste streams make up for the extra costs? A thorough analysis and risk assessment must be conducted.

The mandatory EPR scheme for plastic packaging must make economic sense. Decreasing the overreliance on fossil feedstocks and enabling markets for recycled plastic must have a strong business case in order to be implemented. For further research, several research questions need thorough investigation, for example, what would be the economic valuation for the scheme considering the unique market characteristics in Australia? Or what would be the roles and responsibilities of the EPR scheme? It must consider producers, consumers, local governments, and the state government. Countries around the globe, including Australia, are facing a significant challenge of managing plastic sustainability since China's waste ban, and at the same time, various local initiatives have been initiated to tackle the challenges. Thus, we need to transform the global plastic challenge into an opportunity, and now is the best time to foster decentralised local solutions through innovative social and engineering technologies.

**Author Contributions:** Conceptualization, A.M.C.R. and A.Z.; methodology, A.M.C.R. and A.Z.; formal analysis, A.M.C.R.; writing—original draft preparation, A.M.C.R.; writing—review and editing, A.M.C.R. and A.Z.; supervision, A.Z. All authors have read and agreed to the published version of the manuscript.

**Funding:** This research received no external funding.

**Data Availability Statement:** The data presented in this study are available on request from the corresponding author.

**Acknowledgments:** The authors would like to thank the participants of this research for their involvement and valuable insights.

**Conflicts of Interest:** The authors declare no conflict of interest.

**Glossary**

| | |
|---|---|
| APCO | Australian Packaging Covenant Organisation |
| ARL | Australasian Recycling Label |
| CDS | Container Deposit Scheme |
| CE | Circular Economy |
| CfC | Containers for Change |
| COAG | Council of Australia Governments |
| CoE | Collection efficiency |
| DfE | Design for environment |
| EPR | Extended Producer Responsibility |
| LR | Structured Literature Review |
| MRF | Materials Recovery Facility |
| PET | Polyethylene Terephthalate |
| HDPE | High Density Polyethylene |
| PVC | Polyvinyl Chloride |
| LDPE | Low Density Polyethylene |
| PP | Polypropylene |
| PS | Polystyrene |
| EPS | Expanded Polystyrene |
| RMF | Recycling Modernisation Fund |
| SMRC | Southern Metropolitan Regional Council |
| WA | Western Australia |
| MSW | Municipal Solid Waste |

**Appendix A**

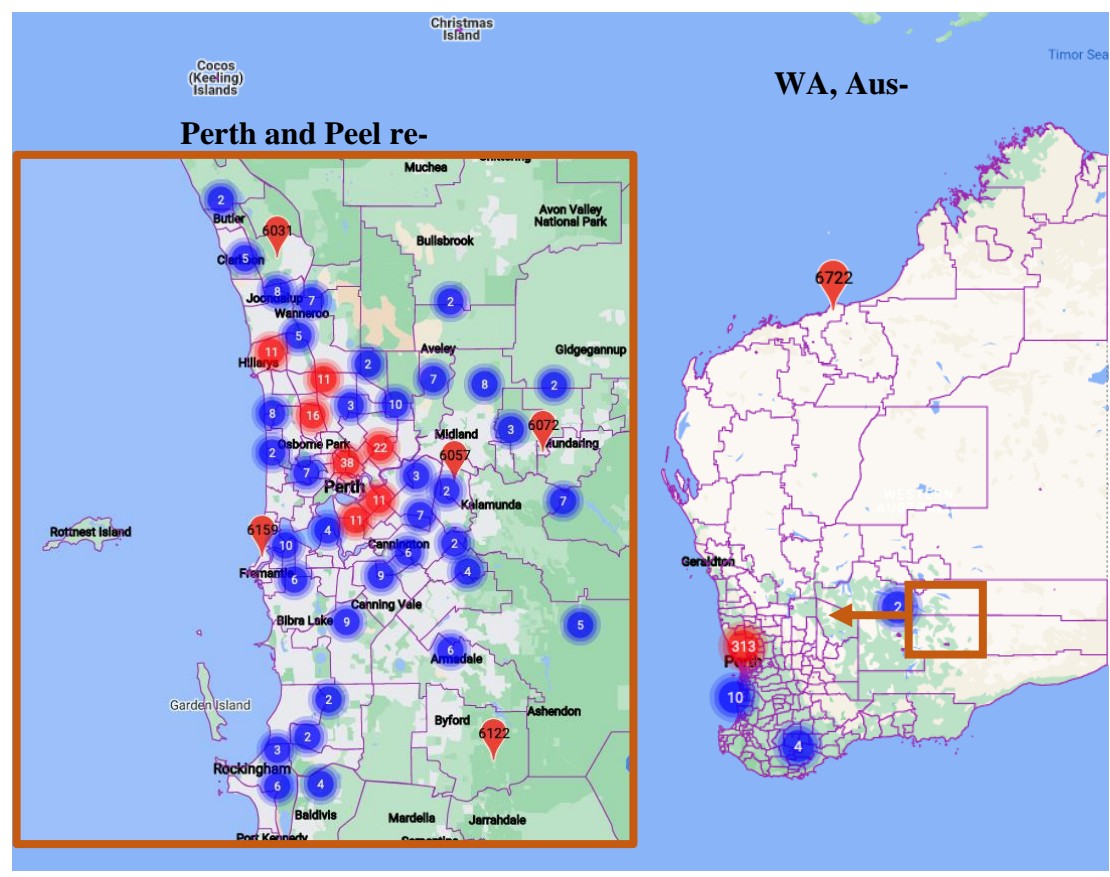

**Figure A1.** Postcodes' distribution from questionnaire's participants.

## Appendix B

**Table A1.** Community participants' background.

| Indicators | Parameters | % | Count |
|---|---|---|---|
| Age group | Gen Z (Below 25 years) | 12% | 43 |
| | Millennials (25–40 years) | 64% | 229 |
| | Gen X (41–56 years) | 19% | 67 |
| | Post War (76+ years) | 0% | 1 |
| | Boomers (57–75 years) | 5% | 18 |
| Highest academic level | Bachelor's degree | 34% | 118 |
| | Secondary education | 28% | 99 |
| | Master's degree | 14% | 51 |
| | Graduate Diploma | 12% | 42 |
| | Graduate Certificate | 8% | 27 |
| | Doctoral Degree | 4% | 13 |
| | Primary education | 1% | 2 |
| Income bracket (Australian dollars) | AUD 0–AUD 18,200 | 14% | 170 |
| | AUD 18,201–AUD 45,000 | 15% | 56 |
| | AUD 45,001–AUD 120,000 | 49% | 51 |
| | AUD 120,001–AUD 180,000 | 16% | 49 |
| | AUD 180,001 and over | 6% | 19 |
| Household type | Apartment/Unit | 18% | 64 |
| | Stand-alone house | 70% | 249 |
| | Townhouse | 8% | 28 |
| | Other, please specify | 4% | 13 |
| Occupational status | Full time | 61% | 218 |
| | Part time | 18% | 64 |
| | Retired | 3% | 10 |
| | Unemployed | 7% | 24 |
| | Other, please specify | 11% | 39 |
| Waste bins in household | One (for landfill) | 1% | 3 |
| | Two (for recycling and landfill) | 55% | 190 |
| | Three (for organic, recycling and landfill) | 41% | 144 |
| | Other | 3% | 11 |

## Appendix C

**Table A2.** Interviews' participants.

| Participant | Affiliation | Job Position | Waste Experience |
|---|---|---|---|
| A | Government organisation | Chief Executive Officer | 23 years |
| B | Environmental organisation | Sustainability officer | 6 years |
| C | Government organisation | Waste and Recycling manager | 18 years |
| D | Business organisation | Administration manager | 18 years |
| E | Business organisation | Administration manager | 18 years |
| F | Environmental organisation | Board Director | 25 years |
| G | Government organisation | Chief Executive Officer | 17 years |
| H | Business organisation | Purchase Manager | 2.5 years |
| I | Business organisation | Managing Director | 4 years |

## Appendix D

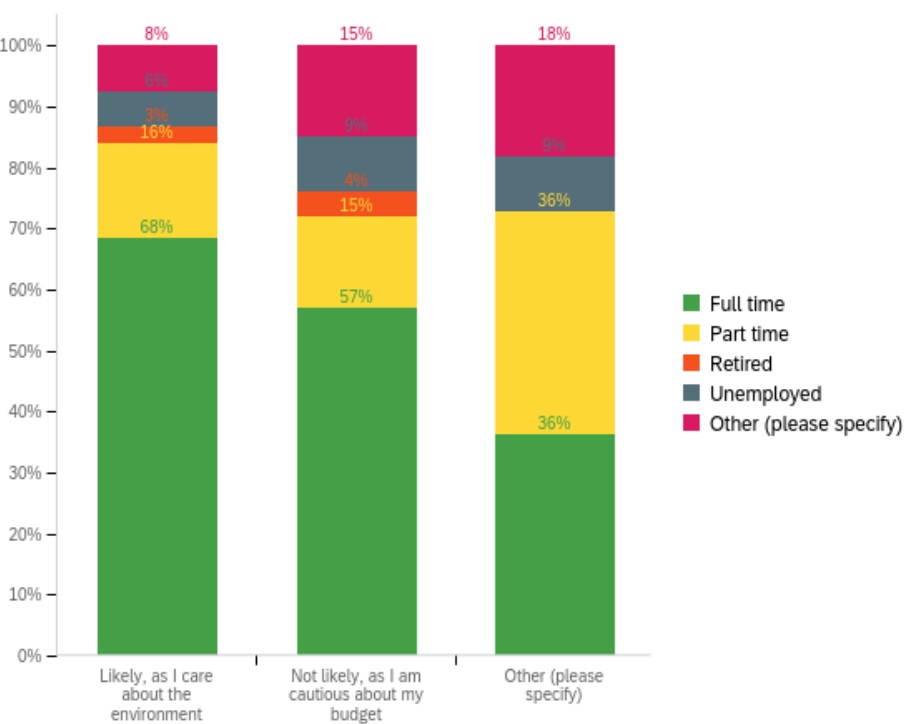

**Figure A2.** Would you buy recycled plastics over virgin plastics available on the market if the price was slightly higher? vs. Occupational status.

## Appendix E

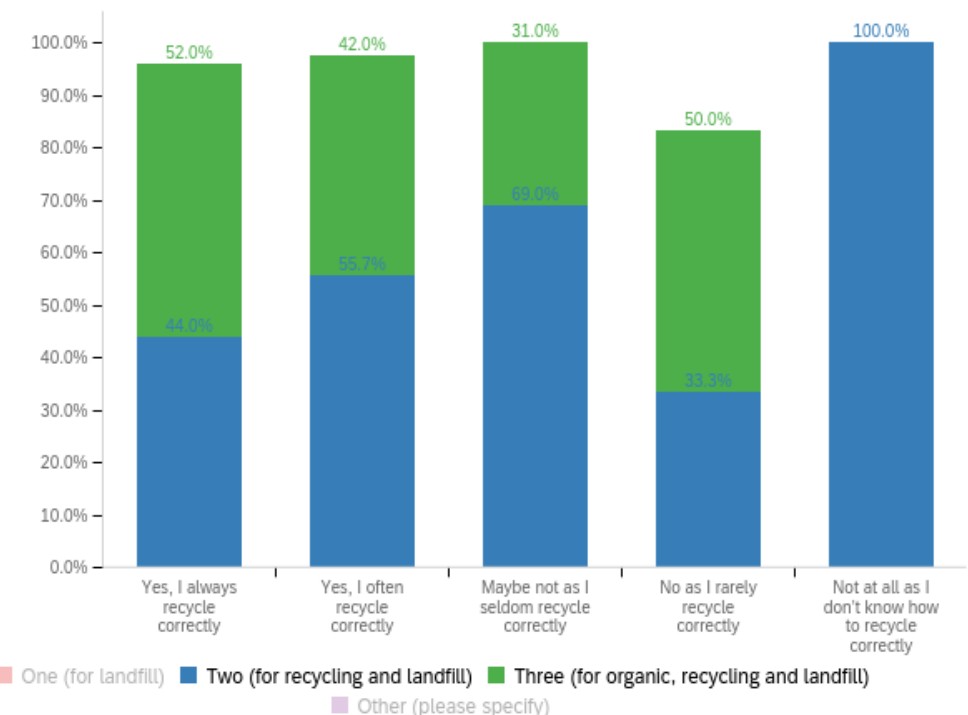

**Figure A3.** Do you think you recycle correctly? vs. Which is your bin system?

**Appendix F**

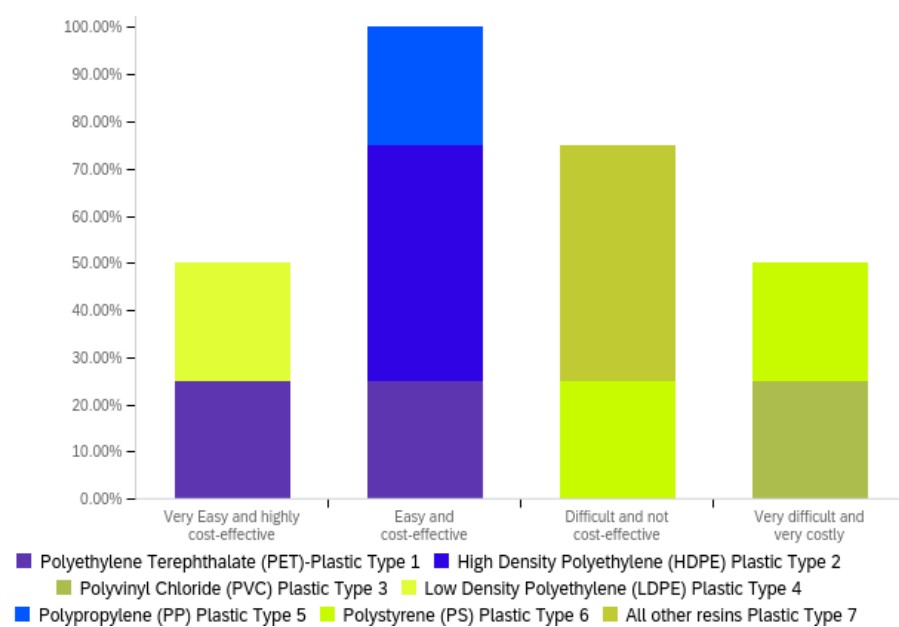

**Figure A4.** How easy and cost effective is the plastic waste you process in your facility?

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
