# Peer review of "The Current State, Challenges, and Opportunities of Recycling Plastics in Western Australia"

_recycling, doi:10.3390/recycling7050064_

Round 1

Reviewer 1 Report

Dear Authors,

The presented work, although not always of very high study depth, is insightful. The information is clear and well arranged. The study design is convincing. Conclusions are sufficiently supported by the results. Results and conclusions make a genuine contribution to better understand the challenges and opportunities related to plastic wate recycling in Australia.

Nevertheless, some issues should find your attention and must be improved:

1. Section 1 (“Introduction”), paragraph 2, sentence 3 (“In 2018-19, 85% of discarded plastics was disposed …”): kindly note the percentages (shares of plastic waste management) indicated in this sentence do not sum up to 100% but to 101%. Please check and update the information.

2. Section 2 must me rearranged and better focused to capture the current plastic waste management situation in Australia. The section currently starts in a too general way, and the first part of this section must be more specific with some relation to the actual topic of this paper, namely plastics in Australia. Furthermore, the situation in Australia with view to current plastic waste management must be covered here in this section in more detail.

3. Section 3: kindly check the beginning of this section. It currently contains two paragraphs which are text from the journal template and not your own work.

4. Section 3.2.3. (“Experts’ Interviews”): Please kindly disclose in appropriate detail the criteria to select the contacted experts and the procedures applied to decide for the individuals who were finally interviewed.

Author Response

Feedback from Reviewer 1:

Dear Authors,

The presented work, although not always of very high study depth, is insightful. The information is clear and well arranged. The study design is convincing. Conclusions are sufficiently supported by the results. Results and conclusions make a genuine contribution to better understand the challenges and opportunities related to plastic wate recycling in Australia.

Authors response:

The authors want to thank the reviewer for finding the article insightful, clear and well-arranged. Moreover, the authors have found your comments valuable to improve the quality of the article further. The authors have considered your feedback and revised the article accordingly.  

Point 1

Section 1 (“Introduction”), paragraph 2, sentence 3 (“In 2018-19, 85% of discarded plastics was disposed …”): kindly note the percentages (shares of plastic waste management) indicated in this sentence do not sum up to 100% but to 101%. Please check and update the information.

Authors response to point 1:

As per the reviewer’s comments, this section has been checked and updated. The source indicates “A little less than 13% (in 2016-17 it was 12%) was recycled and a little less than 3% used for its energy value, mostly in solid recovered fuels for energy recovery.”. The source also indicates that the landfill rate is 85%, therefore this rate was left unchanged in the document. However, the adverb “nearly” was added to modify the recycling and energy recovery percentages. The revised document now reads “In 2018-19, 85% of discarded plastics was disposed of in landfills, nearly 13% was recycled and nearly 3% went into energy recovery”. Before, the word “nearly” was not included.

Point 2 

  1. Section 2 must me rearranged and better focused to capture the current plastic waste management situation in Australia. The section currently starts in a too general way, and the first part of this section must be more specific with some relation to the actual topic of this paper, namely plastics in Australia. Furthermore, the situation in Australia with view to current plastic waste management must be covered here in this section in more detail.

Authors response to point 2:

The authors appreciate the feedback. As per the reviewer’s comments, this section has been rearranged. Further information about the current plastic waste management in Australia was added to the revised manuscript. Information added includes plastic waste flows, type of plastic waste collection systems and plastic waste management policies in Western Australia. Management of plastic waste with EPR schemes was included in a final sub-section. This subsection has been included to show the global tendency towards a more circular plastic waste management.

Point 3

  1. Section 3: kindly check the beginning of this section. It currently contains two paragraphs which are text from the journal template and not your own work.

Authors response to point 3:

Thank you for the feedback. The beginning of section 3 was removed as it did not belong to the article.

Point 4

  1. Section 3.2.3. (“Experts’ Interviews”): Please kindly disclose in appropriate detail the criteria to select the contacted experts and the procedures applied to decide for the individuals who were finally interviewed.

Authors response to point 4:

Thank you for the feedback. Section 3.2.3 has been modified so it can clearly state the expert selection criteria.

Reviewer 2 Report

Manuscript title ''The Current State, Challenges, and Opportunities of Recycling Plastics in Western Australia'' I think overall presented well, however I think article still need some improvement in terms of research synthesis and conclusion and suggestion for future research in this area. 

Author Response

Feedback from Reviewer 2:

Manuscript title ''The Current State, Challenges, and Opportunities of Recycling Plastics in Western Australia'' I think overall presented well, however I think article still need some improvement in terms of research synthesis and conclusion and suggestion for future research in this area. 

Authors response:

The authors want to thank the reviewer for finding the submission a well-presented article. Moreover, the authors have found your comments valuable and have considered your feedback and revised various sections of the article (including methodology, findings, conclusion and further recommendations) to improve the quality of the article.